# Is Corporate Political Activity an Investment or Agency? An Application of System GMM Approach

**Woon Leong Lin** 

Faculty of Economics and Management, Universiti Putra Malaysia, UPM, Serdang 43400, Malaysia;
linwoonleong@gmail.com

**Abstract:** Corporate political activity (CPA) has been recognized as bearing a significant impact on financial performance (FP). Nevertheless, there has been a lack of considerable research to date. The results of the research regarding the relationship between CPA and FP have been contradictory and this has necessitated further investigation of this relationship. Nonetheless, rather than examining the relationship between CPA and FP, research scholars have revealed that a contingency perspective must be employed for revealing the conditions and the context which enhance the relationship between these two constructs. This study offers a quite distinctive viewpoint with respect to the link between CPA and FP as regards the corporate reputation perspective. For this reason, the study obtained data from the Fortune list of top 100 World Most Admired Companies (WMAC) for the period of 2007 and 2016. This data was utilized to examine the relationship between CPA and FP using the dynamic panel data system GMM (Generalized Method of Moments) estimator. This study finds virtually no support for the hypothesis that lobbying and PACs (political action committees) represent an investment in political capital. Instead, CPA is symptomatic of agency problems within firms. This study also argues that corporate reputation moderates the effect of CPA on the FP and the analysis supports the argument. Our results are particularly useful in light of the reputable corporation, which is greatly to likely increase the use of corporate funds for political contributions.

**Keywords:** corporate political activity; firm performance; corporate reputation; system GMM

## 1. Introduction

The CPAs (which stand for corporate political activities) have been extensively used by many companies to enhance their performance, considered their administrative area and also influence the governmental regulations (Hillman et al. 2004). This increased concern of the CPA is quite remarkable, and, in the previous 20 years, it was viewed that the firms have considerably increased their CPAs (Kerr et al. 2014; Drutman 2015; Lin 2019). Many firms can successfully influence the governmental executives by direct petitioning, voluntary treaties, campaign contributions, PACs (political action committees), the inclusion of the government in the boards of the companies or bribery (Hansen and Mitchell 2000; Delmas and Montes-Sancho 2010; Sun et al. 2012).

Many CPA-related reviews have depicted the way in which the private firms engage and tackle completion in the non-market movements (Baron 2001). Certain researchers claimed that the non-market actions are employed for soliciting some special benefits from the authorities of the government. Such a standpoint can be employed as a distinctive research topic, as it portrays the performance of the firm in the political domains (Hillman et al. 2004). Although many empirical study reports have facilitated in developing an improved understanding of the CPA, an inclusive discrepancy is still present in this literature due to two primary reasons. First of all, not many studies attempted to resolve the specific cause and the CPA findings and the published studies diverged from their chief objectives. Thus, this phenomenon was examined in many ways; secondly, several researchers

mentioned that the previous studies discarded the major parameters that regulated the relationship between CPA and FP, which caused the misspecification of all the models. Hence, this study resolved this matter by regarding the corporate reputation (CP) to be a moderating parameter and setting up a boundary condition as regards the correlation between CPA and FP. Thirdly, prior literature and studies have ignored the endogeneity of the relationship between CPA and firm performance. Ignoring both unobservable heterogeneity (as in the pooled OLS model) and the dynamic CPA–FP relationship (as in the fixed-effects model) would lead to biased estimates as it has been shown in previous studies.

This paper aims to give an insight into this significant issue by determining whether this proclamation is well supported. It is done by assessing the CPA of a vast amount of companies that are recognized globally as most admired firms as per the well-known annual survey by Fortune, in relation to that of a control sample for a period of 10 years. This study assess WMAC firm performance by looking at the impact of CPA on FP and even the moderating effect of CP on the relationship between CPA and FP. This study used the information from the list of Fortune's WMAC firms for a period of 10 years (2007–2016) and determined whether the CPA was beneficial to the stakeholders in the firm. This study also established whether the corporate reputation moderates the relationship between CPA and FP. The finding showed no significant relationship between CPA and FP. Also, this neutral relationship could also be partly because of the fact that CPA of a firm is not able to provide concrete benefits or even facilitate the firm to acquire more governmental deals and increase the possibility of getting the bill approved in the congress. It suggested that the expenditure of the agency (i.e., an ineffective corporate funds utilization) for the lobbying movements controlled the strategic advantages of the lobbying. Secondly, it was also observed that there was a negative moderating effect of CPA and CP on the FP, wherein CP functions as a moderator of the CPA effects on performance; Hence, this study posited that CPA as a non-market policy is more effective in those firms having low CP compared to those with high CP.

This research played two-fold roles. First of all, this study assessed the non-market movements used by the firms. The opportunities varied from active and the more precise (like CPA) to the representative and the less precise (like CP) (Nye 2004). This study has used traditional CPA frameworks (Hansen and Koehler 2005) and CP-related managerial studies (Anastasiadis 2014; Rehbein and Schuler 2015). The CP involves many processes, strategies and activities employed by the firms for fulfilling the societal and political expectations (McWilliams and Siegel 2001). This study aspires to contribute to the current literature as regards the parameters that inspire the firms and persuade them to take part in the activities beneficial for the community (Bénabou and Tirole 2010). Additionally, this study also examines how and whether the firms can perform well by doing lobbying and PACs, where their moral conduct depends on the staff members, customers, financiers and their stakeholders (Margolis et al. 2009).

Another thing is that this study offered new empirical findings which without any endogeneity issues. As shown by previous reports, CPA, FP and CP were interdependent (Hermalin and Weisbach 1998). The previous studies did not control the endogeneity and thus it is possible that their findings are biased (Wang 2015). Any endogeneity, present in the regression models, characterizes the condition in which the explanatory (or endogenous, i.e., FP) parameters are related with the error terms or both the terms of error are correlated in the studies of structural equation modelling. Some scholars (Arellano and Bond 1991; Blundell and Bond 1998) developed and presented a GMM (generalized method of moments) model, which was employed for the dynamic panel data system, in which the cause-and-effect relationship between the fundamental phenomena was viewed as dynamic over time.

The remainder of this article is organized as follows. Section 2 includes a literature review. In Section 3, the theoretic framework and hypotheses development have been outlined. Section 4 elucidates all the techniques, estimation approaches and the variable descriptions. All results have been provided in Section 5. Section 6 outlines the robustness test; all results as well as the research prospects are discussed and finally concluded in Section 7.

## 2. Literature Review

### 2.1. Corporate Political Activities

CPA involves intermediate activities which are present between the traditional businesses and a public context. The CPA involves "any distinguished goings-on of the firm which can influence governmental policies or procedures" (Getz 1997, p. 32). On a strategic plane, CPA involves some activities which are performed in a non-market situation for developing value by improving financial performance (Baron 2001). These descriptions act as a baseline for the expert and administrative studies that seek to improve the insight regarding CPA's nature. Here, CPA has been defined as the efforts made by the firm to solicit certain governmental policies which also profit the firm (Hillman et al. 2004). The reality of all the companies which are occupied in the political procedures is very discouraging as each firm is impacted by the governmental regulations and policies (Stigler 1971).

It was observed that the firm performance relies on the managers' capability to maneuver all financial and political markets (Schuler et al. 2002). Many studies concerned with CPA have regarded the political strategies of the firm, together with their differential strategies, as ones which can incite a desirable impact on policy interventions (Oliver and Holzinger 2008). The CPA can increase the organization's competence and has been categorized as a non-market policy (Chen et al. 2015). Usually, the CPA was defined as the information transmission that takes place in private locations and meetings involving all politicians and their groups of interest, agents or staff. This information can be transmitted in the form of messages, statistics, forecasts, facts, commitments, threats, arguments, signals or a combination of all these different forms. The interest groups normally have their private financial plan and are more prone to investing in these actions; nonetheless, the handover to the politicians is usually not explicit (in comparison to the campaign contributions) (De Figueiredo and de Figueiredo 2002).

The key objective of the CPA is improving the organizational performance (Schuler et al. 2002). Thus, it helps the firms, with CPA access, to reduce their uncertainties (Hillman and Hitt 1999), get political assistance (Keim and Zeithaml 1986; Keim 2001) and also strengthens them so as to implore certain competitive advantage (Insead and Chatain 2008). Most of the companies perform political activity for reducing the risks of exposure (Frynas and Mellahi 2003; Keillor et al. 2005), especially in the emerging nations that have to deal with higher political threats, which occur due to poor governmental regulations and certain institutional loopholes (Henisz and Zelner 2003; Rajwani and Liedong 2015).

### 2.2. Relationship between CPA and FP

As per the literature findings, establishments that are politically well connected enjoy superior performance compared to the firms having few political connections. For instance, systematic meta-analyses (Lux et al. 2011) have demonstrated that political tactics cast a positive impact on the firm performance as they facilitate the firms to obtain beneficial regulatory conditions (Antia et al. 2013) in addition to an access to the monetary resources (Khwaja and Mian 2005), which brings about an enhancement in the value of the firm (Johnson and Mitton 2003; Roberts 1990). This result is backed by Richter et al. (2009), who states that the US firms that lobby more in a specific year have been discovered to pay lesser effective tax rates in the following year. A similar finding for China has been suggested by Wu et al. (2012) wherein managers have to pay lesser taxes for the firms that are politically linked. Moreover, these companies have exclusive access to financial resources (Boubakri et al. 2012; Unsal et al. 2016), to enjoy extended duration of debt maturities (Boubakri et al. 2012) and are substantially leveraged (Fraser et al. 2006). Similarly, campaign contributions can be positively related to the abnormal future returns (Cooper et al. 2010) in addition to the political links that can be positively related to the returns on assets (Imai 2006). Thus, CPA has viewed as an investment to the company.

Majority of the literature has mentioned a positive CPA-FP correlation Nevertheless, some studies have demonstrated contradictory evidence. A small number of research studies have mentioned that poor corporate administration can also surface from CPAs, thus causing poor firm performance (e.g.,

Fan et al. 2007) or reduced exporting activities (Lee and Weng 2013). Aggarwal et al. (2012) found out that 'soft money' donations can be negatively related to that of the returns. Similarly, Hadani and Schuler (2013) stated that a negative impact is cast by CPA on the firm's market worth. Politically linked firms show inferior accounting performance after privatization (Boubakri et al. 2012) as well as after IPOs (Fan et al. 2007), show poor growth (Chaney et al. 2011) and charge higher loan interest rates (Bliss and Gul 2012). A study conducted in Italy has demonstrated that politicians who are also bank directors exert a negative effect on the revenues from interest, quality of loan and on the level of capitalization (Carretta et al. 2012). Thus, firms with political links might not enhance the value of a firm, which is possibly traceable to the costs and the limitations associated with political connections (Okhmatovskiy 2010). From these points of view, CPA is an agency cost to the company.

Several reviews do not find any substantial relationship among CPA and FP. Ansolabehere et al. (2004) discovered no advantages of campaign donations, whereas Hersch et al. (2008) claim that campaign contributions do not generate any financial resources. Three studies quantified multiple results and reported mixed results. As per Faccio et al. (2006) and Lin (2019), though politically linked firms are able to affect the bailout policies of the government to favor them, they show poor effective performance. In the same way, Tu et al. (2013) stated that while politically connected firms pay a lesser premium during privatization, they report lower stock and operating performance after privatization.

To sum up, the outcomes of the earlier studies regarding CPA-FP correlations are diverse. Thus, there is incomplete and uncertain information on the impact of CPA on FP. This literature gap needs an academic scholarship to focus on empirically confirming these tenets (Oliver and Holzinger 2008; Rudy and Cavich 2017).

### 2.3. Concept of Corporate Reputation

The research for corporate reputation (CP) primarily had to deal with the huge differences among the several definitions of the idea of "corporate reputation." Even though still there are some differences, several authors have attempted to provide an integrative description of recent formulations. Gotsi and Wilson (2001), in their past and present literature reviews regarding various domains, determined the following description of corporate reputation: "a stakeholder's overall valuation of a firm over time. This valuation is done on the basis of the stakeholder's direct encounters with the firm, any other mode of symbolism and communication that provides knowledge about the firm's activities and/or contrasting these actions with those of the prominent rivals" (Gotsi and Wilson 2001, p. 25). This study suggests considering the fact that it is viable via communication to substitute the direct experiences of the individuals with surrogate experiences and therefore facilitate a public reputation to exist (Mahon 2002; Dozier 1993). This observation is consistent with Fishbein's view of attitudes being eventually "attained from direct encounters with objects and from interaction pertaining to them with other sources" (Loudon and Della-Bitta 1993). Note that this idea explicitly allows the reputation to change within different groups of stakeholders, although in this paper's empirical portion, this study will claim that the impacts of reputation are rather comparable between various stakeholders. Reputation is associated with but distinct from, the image of the firm because it is a more robust notion as regards company's time and efforts for communication.

Thus, the idea is more intimately connected with a firm's "personality" than the (communicated) perception. Schwaiger (2004) believed reputation to be related to attitudes, involving cognitive as well as effective components and defined solely by denotative elements. This is opposing the customary knowledge in image studies, which describes the image as mainly comprising connotative elements (Gensch 1978). This definition appears to be very effective and thus it will be accepted for this paper. An important legitimacy problem in previous research regarding reputation has constantly been that its multi-dimensionality has not been in line with the appropriate conceptualization. This analysis is especially applicable to the Fortune "Most Admired" indicators as devised by Fryxell and Wang (1994). In the study of WMAC, an overall reputation score is obtained as the mean of nine parameters rated

by scholars from within the firm's domain on 11-point scales). The WMAC includes the 500 largest firms worldwide.

## 3. Research Hypotheses

Based on the previous divergent viewpoints regarding CP and an absence of studies that examined the interaction effect between the CPA and FP, this study has determined whether the organization's involvement in the CPA and CP can enhance its performance. This framework has been depicted in Figure 1.

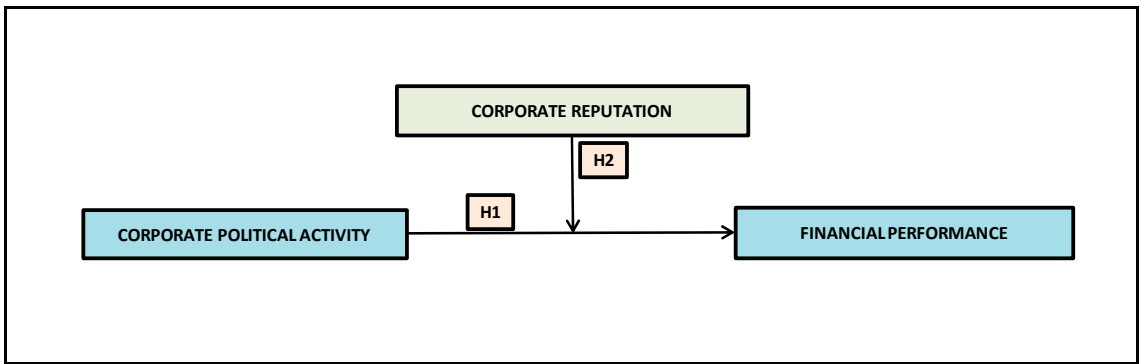

**Figure 1.** Theoretical Framework.

### 3.1. Effect of the CPA on FP

In this research, this study has explored the concept of organizational behavior, which explains that the CPA is a significant element of the zero-sum political game that cannot be perfectly assessed. This suggested that the CPA cannot significantly improve the FP (Hadani and Schuler 2013; Hadani et al. 2017; Lin 2019).

Moreover, the theory of organizational behavior is based on the concept which suggests the presence of a neutral correlation between the CPA and FP. It has been believed that many executives possess certain cognitive limitations and preconceptions, which are related with the organizational customs and constitution and are present in the firms that imperfectly screen, scan or process the external information (Cyert and March 1963). The implementation of the CPA is complemented by several uncertainties, institutional limitations, time pressure, asymmetry of information and is also characterized by many actors together with a considerable causal ambiguity among the activities and the outcomes (Hart 2004). As compared to the other market techniques that include rapid and comprehensible feedback, CPA has slower feedback, as regards the spending and a quite ambiguous relationship between the cause and the effect (Hart 2004).

Moreover, for handling uncertainties, several firms formulate some patterns for decision-making, known as the SOPs (standard operating procedures), which involve a set of actions used for specific situations (Den Hond et al. 2014). Lobbying a particular Congress member and also contributing financially to the campaigning is regarded as SOPs, which involves information sharing with the government officials and supporting the voted representatives (Den Hond et al. 2014). Over time, these customs are established, handled and standardized until it is observed that these actions cause a negative outcome (Den Hond et al. 2014; Snyder 1992). Due to the ambiguity of all political procedures and the vagueness associated with the estimation of benefits and costs of CPAs (Kroszner and Stratmann 2005); there is a small probability that most of the SOPs would be investigated with respect to their impact on FP (Hart 2004). On the basis of the organizational theory, the behavioral standpoint of the firms suggests that they must face hardships while evaluating the CPA funding. Nevertheless, they can cooperate because of the zero-sum reimbursement of CPA and the demanders of policies. Thus, this study expected that there would be a lack of a positive relationship between the CPA and FP.

**Hypothesis 1.** *CPAs represented by the lobbying expenditures and the political action committee (PAC) contributions exhibit no significant relationship with the FP.*

*3.2. Effect of the Interaction between CPA and CP on FP*

Only a few studies have performed research regarding the CPA-CP relationship, in spite of its significance (Den Hond et al. 2014). This relationship has to be disentangled for understanding all the inferences of the relations between CPA and CP. Lyon and Maxwell (2008) reported that this kind of research must be performed as they considered that CPA has to be integrated into the entire framework of CP. In this research, I predicted that CP shows a constructive and regulating influence on the CPA-FP correlation because certain policies developed by the companies can encourage CPA. The political connection is a significant resource which can definitely provide a competitive advantage to the firm in the political domain or in the market (Bonardi 2011). CPA is necessary for forging strategic affiliations and partnerships with the lawmakers and the law enforcers (Schuler et al. 2002). It was also viewed as an important factor which was employed for exercising authority in the different legislations and regions like the EU (Bouwen 2002). The CPA is a limited resource. Thus, its implementation can be tricky and can cause policies which necessitate some kind of return. The real nature of these 'access goods' (Bouwen 2002) varies across all legislation, officials and administrators. For instance, the EU executives require in-depth knowledge and facts about the different needs or interests of various industries (Bouwen 2002).

More prospects are proposed for an indirect approach on the basis of the relationship among the CP firms and their stakeholders, like the NGOs or the community. Additionally, based on the direct approach, which the firms have to the administrators and the politicians, the social-accountable organizations 'pooseback' on their political supporters and subsequently earn benefits and opportunities through an indirect approach (Den Hond et al. 2014). The basic source of disagreement with reference to the communal network is the fact that agents are used for getting in touch with other political factions (Burt 2009). For instance, network research was conducted in the dairy industry of the Netherlands, which confirmed that the high levels of stakeholder management having a greater CP can enhance the relationships and facilitate the better development and stronger relationships with the government officials (Mathis 2007). As compared to the low CP organizations, the high CP firms have more varied and strong relationships with the community and NGOs. Such associations enhance the effectiveness of the CP organizations in the political domains; since they provide greater clout and facilitate the firms promote better positions with respect to certain issues. The provision of information is regarded to be a CPA policy and the position of the organization regarding these issues is represented to the officials and the administrators. Certain studies (Rehbein et al. 2005; Rehbein and Schuler 2015) mentioned that the firms generally display superior status in their interaction with the non-business groups. Usually, the cooperative enterprises result in the status that shows a greater knowledge and extent and present a transparent picture of the benefits and costs as compared to the firms which are dependent on their employees. Hence, this study presented the second hypothesis as follows:

**Hypothesis 2.** *CPA positively impacts the FP in firms with a higher CP, as against its impact in the low CP firms.*

## 4. Empirical Model, Methodology and Data

*4.1. Measurement of Corporate Political Activity*

This study measured CPA as some organizational non-market operations that consist of any one of the given operations: PAC donations and lobbying expenses (Hadani et al. 2017). The lobbying expenditure records and the PAC were acquired from the CRP (Centre for Responsive Politics) at the US Senate website, which has been giving organizational expenses on lobbying operations on a tri-monthly

basis since 1999. The 1995 Lobbying Disclosure Act instituted listing and filing requirements for firms seeking to affect US government strategies and/or implementation of the Federal program. Every registrant is required to file six-monthly publications within 30 days prior to the end of the semi-annual duration, with the aim to report their lobbying endeavors as well as the expenses involved. This study linked the data with DataStream in order to extract only public-sector firms. As the CRP data does not provide company identifiers, this study had to physically verify these organizations' names by matching data of CRP and DataStream. The main drawback of this procedure is that it was not possible to trace the quantities that the firms spent on specific bills. To make sure that either financial datasets or the CPA are related to the same period, this study discarded every firm that presented a non-calendar financial year-end, which minimized the amount of the sample firms to 1294 observations, after investigating all the dependent parameters that were analyzed.

## 4.2. Measurement of Corporate Reputation

Information regarding the firm's reputation was obtained from the yearly review of corporate reputations conducted by Fortune magazine. Fortune has performed the review for the last 11 years and published results of the summary in the January/February edition known as "World's Most Admired Companies"[1]. Over 8000 administrators and external industry experts are requested to rate firms within their domain on nine attributes: financial health, wise use of assets, long-term value of investment, organizational quality, product and service quality, innovation, worldwide competitiveness, capacity to attract, train and retain employees and societal and environmental obligations. The Fortune review was chosen for many reasons. First of all, it gives comparable data for a long time period for a huge amount of organizations in 32 different nations. Firms are added to and discarded from the dataset eventually because of mergers and performance changes but the amount is rather stable. Another reason is that the respondents' quality is comparable to people who could be acquired elsewhere because respondents only rate organizations familiar to them. They can directly access the industry data which is salient to analyze the reputation of a firm. Third, McGuire et al. (1988), Gatewood et al. (1993), Fombrun and Shanley (1990), Fryxell and Wang (1994), Aguilera-Caracuel and Guerrero-Villegas (2018) and Lin et al. (2019) have utilized the instrument for evaluation of the reputation of the firm. This allows us to connect our findings to a wider literature body. The sampling characterizes the Fortune 1000 list of big corporations that are placed by sales in about 70 industries. The ranking is carried out each Fall since 1982, having the summary results announced every January which covers the largest firms in about 25 trade groups, with ratings for the highest 10 in every group. For each firm-year observation, CP is ranked throughout a period that ranges from 0 to 10 in scale. The Fortune survey grades each firm compared with its respective prominent competitors based on 9 traits, through an 11-point scale, wherein 0 is poor while 10 is excellent.

## 4.3. Measurement of Financial Performance

Measures based on accounting are generally known to represent historical and short-term financial performances. The supporters of performance measurement claim that market-based pointers can be affected by several factors that are not related to political operations. Research scholars agree on the lesser neutrality that measures based on accounting represent and thus focus on the significance of measurement based on values that links the investor to the interests of the shareholders (Brammer and Millington 2008). Conversely, Griffin and Mahon (1997) concentrate on the significance of implementing traditional accounting-based measurement, as newer appraisals based on values can uncover more than just financial performance. In spite of the financial pointers' limitations, they have been extensively known to provide the most precise depictions of corporate financial performance (Simpson and Kohers 2002). Also, these results from previous research do sustain the claim that

---

[1] See detail of methodology, please visit: http://fortune.com/worlds-most-admired-companies/list/.

CPA is more liable to demonstrate stronger accounting-base correlational links as against returns related to returns (Orlitzky et al. 2003; Peloza 2009). Provided these claims, this study resolved to use alternatives for corporate performances in this paper, wherein the three measures of performance based on accounting are key ROE and ROA. ROE functions as robustness test.

ROA refers to the main parameter in this research; it helps in measuring the extent of the net income after paying tax and the interest prior to the inclusions of ideal dividends, on the basis of the per-unit mean previous-year as well as existing assets. Hull and Rothenberg (2008, p. 785) mentioned that ROA "refers to corporate productivity with regards to the total set of assets and resources under control." This quantity is widely used, as observed in the reports that follow: (Bliss and Gul 2012; Richter et al. 2009). This study used ROE (Return on Equity) as a test of our robustness since ROE also is probably the most widely known among productivity measures (Hawkins 1998) and of the highest interest to shareholders (Bernstein and Wild 1989). It refers to the profits that stockholders/investors are achieving on their investments.

## 4.4. Control Variables

All the key control parameters are important to our capability to reflect the marginal effect of political actions to each dependent parameter in this study. At the organizational level, this study regulated for many items that presented a consistent relation with corporate productivity, including company volume in terms of fixed assets and revenues, free currency, corporate influence and the amount of advertising. The scale of these firms is measured by overall sales; fixed benefits are reflected in the balance sheet assets sum. As per Dang et al. (2018), proxies of different size capture different viewpoints of "firm size" and therefore have different inferences in corporate finance. For instance, total assets are a measure of the firm's total assets, whereas total sales are more associated with product market and are not progressive. The choice of the measures of the firm size can be a hypothetical and empirical concern. For instance, if researcher scholars want to regulate the "size" of the company in its market, then they must use total sales; if they wish to regulate the stock market size, then they must use market cap; if the size represents the total assets from which the firm can generate returns, then they must use total assets. The free flow of cash (to equity) is computed as net incomes with depreciation accounts, lest capital expenses and alterations to non-cash working assets, net debt concerns and preferred payments. The corporate influence is represented by the current ratio, which is calculated by breaking up a firm's current assets into current liabilities. Lastly, advertising spending (to sales) must be included in assessments, since these have also been confirmed to affect CPA and FP (Lev et al. 2010).

## 4.5. Empirical Model

A two-step method system GMM (Windmeijer 2005) was employed to estimate the impacts of complementarity and substitutability on the FP and CPA and CP's moderating impact on the relationship between the FP and CPA. A GMM system was employed to estimate the impact of CPA on FPs, as well as develop certain dynamic dimensions in the dataset. Post this, the moderating impact of CP activities on the relationship between CPA-FP was studied. The model described earlier (Waisman et al. 2015; Brown 2016) was referred to when designing this empirical model. After analyzing all the previous reports, general practice was seen to develop, in which the empirical relationship between the FP and the CPA pertaining to the organizations was measured as follows:

$$\text{Financial performance} = \int \text{Corporate political activity} + \int \text{Control } \mu_i + \varepsilon_{it} \qquad (1)$$

The Equation (1) was also written as:

$$\text{ROA}_{it} = \alpha + \beta_1 \text{ROA}_{it-1} + \beta_2 \text{CPA}_{it} + \beta_2 \text{Control}_{it} + \mu_i + \varepsilon_{it} \qquad (2)$$

Here, CPA signifies the lobbying expenditure and PAC contribution, ROA denotes the Return on Assets, $\varepsilon_{it}$ defines the residual or disturbance term, $\mu_i$ indicates to the unobserved firm-specific effects term, t signifies indexes years and i denotes indexes firms. These variables can be associated with those that are generally mentioned in various corporate political surveys, such as free cash ratios, revenues, total assets, (a natural logarithmic technique is used to translate all these elements), corporate leverage and advertisement intensities. To test the hypothesis stating CP to act as a moderating factor, which can cast an impact on the relationship between FP and CPA, an extension of Equation (2) is done to combine all the interactions that occur between CP and CPA. The model specifications make use of these interaction terms (CPA*CP) (Brambor et al. 2006) in the following way:

$$ROA_{it} = \alpha + \beta_1 ROA_{it-1} + \beta_2 CPA_{it} + \beta_3 CP_{it} + \beta_4 (CPA*CP)_{it} + \beta_5 Control_{it} + \mu_i + \varepsilon_{it} \qquad (3)$$

*4.6. Econometric Methodology*

A well-known econometric technique is the Generalized Method of Moments (GMM) employed for the determination of the dynamic panel models, as it relies on lagged variables. In this study, the GMM panel estimator process is used as put forward by Holtz-Eakin et al. (1988) but others (Arellano and Bond 1991[2]; Blundell and Bond 1998) have extended it because of two reasons. First, there is a need to control the firm-specific impacts, which is not possible with organization-specific dummies because of the dynamic structure associated with all formulated regressions. Second, all simultaneity biases, which may occur due to the likelihood of some of the explanatory variables being endogenous, can be controlled with this technique.

In this research work, a two-step system-GMM estimator[3] was employed to analyze the impact of CPA on FP, by observing the interaction between CP and CPA. The GMM estimators' consistency relied on three specification tests, that is, the Hansen test of the over-identifying restrictions, difference-in-Hansen test for multiple instruments and the autocorrelation test of disturbances (Arellano and Bond 1991). For the Hansen test, a failure in the rejection of the null hypothesis points towards all instruments' validity as well as accurate model specification. For difference-in-Hansen test, the failure in rejecting the hypothesis signifies that there are multiple instruments or there is a lack in proliferation for the instrument. For the autocorrelation test, an absence in the second autocorrelation should not be rejected (AR2).

*4.7. Data and Sample Period*

Our predictions were tested in the context of business by employing a sample with 100 publicly traded Fortune's World Most Admired Companies (WMAC) for 2007 to 2016. To deal with sampling selection problems, a balanced panel was not mandated, thus there is a difference in the sampling numbers with the year. Our estimation method used all the available observations possible. Moreover, to construct the dimensional dynamics for the database by the addition of lagged values for the dependent variable, this study included corporations with no less than seven consecutive years, while excluding those firms which did not present full information. This allows the study to estimate whether this study could consider employing two types of non-market strategies to complement or substitute and how this would impact the FP due to the interaction effect. This study's dataset is an unbalanced panel involving firms from years 2007 to 2016. At most, the dataset includes 100 firms in the cross-section and consisted a total of 1294 firm-year observations. To construct this, data found in (i) public records regarding the lobbying and PACs expenditures by the firms (ii) the firms' accounting statements of and (iii) the most complete database accessed on CP were combined.

---

[2]  See (Arellano and Bond 1991; Blundell and Bond 1998) for more details about the estimation procedure used in the initial System GMM estimation.

[3]  Estimations were carried out using the Stata module Xtabond2 developed by D. Roodman (2009).

## 5. Empirical Results and Discussion

Tables 1 and 2 lists out the descriptive statistics pertaining to all variables employed in this study. The relationship for most of the variables was seen as being statistically significant and matched with the study's expectations. This was indicated by the control variables, which were linked to other variables. Moreover, the correlation matrix indicates that multicollinearity issues were not associated with the study's variables as a value beyond 0.70 was not recorded. Even though particularly high pairwise correlations were not seen amongst independent variables, the firm revenue could be correlated with other variables, which could result in a multicollinearity issue. After an in-depth evaluation of the variance inflation factor (VIF) pertaining to variables, there were no serious multicollinearity concerns observed. In this study, a maximum VIF value of 2.24 and a mean VIF value of 1.56 were employed. When the variance inflation factors (VIF) were analyzed, no multicollinearity issues were observed in the data, as the VIF values were far versus the threshold value of 10.

**Table 1.** Descriptive statistics.

| Variable | Unit of Measurement | Obs | Mean | Std. Dev | Min | Max |
|---|---|---|---|---|---|---|
| ROA | Net income before extraordinary items to a total asset of the firm | 1000 | 6.001 | 5.841 | −30.491 | 40.270 |
| ROE | Net income before extraordinary items to total equity of firm | 1000 | 12.357 | 33.223 | −201.090 | 227.880 |
| ln CPA | Log of the total amount of lobbying to total net sales of the firm | 1000 | 8.429 | 1.834 | −2.323 | 11.635 |
| CP | Reputation scores from 0 to 10 | 1000 | 7.789 | 0.702 | 5.210 | 9.800 |
| Leverage | Long-term debt of firms to their total equity | 1000 | 5.267 | 28.596 | 2.120 | 91.210 |
| Free Cash | Free cash flow to the total of sales | 1000 | 8.510 | 16.558 | −218.360 | 241.110 |
| Advertising | Advertising expenses to total net sales of firm | 1000 | 11.284 | 14.107 | 0.268 | 108.480 |
| ln Total Assets | Log of total asset of the firm | 1000 | 11.470 | 1.267 | 7.382 | 12.632 |
| ln Revenue | Log of total net sales of firm | 1000 | 11.157 | 1.206 | 4.121 | 13.095 |

Notes: All statistics are based on original data values.

**Table 2.** Correlation.

| | 1 | 2 | 3 | 4 | 5 | 6 | 7 | 8 | 9 |
|---|---|---|---|---|---|---|---|---|---|
| ROA | 1 | | | | | | | | |
| ROE | 0.5158 | 1 | | | | | | | |
| ln CPA | 0.1429 | 0.1391 | 1 | | | | | | |
| CP | 0.2546 | 0.0469 | 0.1841 | 1 | | | | | |
| Leverage | −0.0414 | 0.5407 | 0.0643 | −0.0287 | 1 | | | | |
| Free cash | 0.2115 | 0.0540 | 0.0208 | 0.1137 | −0.0301 | 1 | | | |
| Advertising | 0.0487 | 0.0182 | −0.0384 | −0.0246 | 0.0503 | 0.0128 | 1 | | |
| ln Total Assets | −0.1435 | −0.0378 | 0.3271 | 0.0753 | 0.0413 | 0.1248 | 0.0225 | 1 | |
| ln Revenue | 0.0290 | 0.0648 | 0.3257 | 0.1687 | 0.0808 | −0.0509 | −0.2130 | 0.6866 | 1 |

Notes: All statistics are based on original data values.

### 5.1. CPA-FP Relationship

The static and dynamic OLS, fixed effects models, as well as the estimation of ROA with the system GMM, are presented in Table 3. This article employed CPA to evaluate the impact of ignoring the dynamic CPA–FP heterogeneity as well as nexus. The misspecification tests (the second order serial correlation test, i.e., AR 2 and the Hansen test, to estimate other restrictions) were also validated. This authentication approach allowed us to validate the GMM model specifications' appropriateness. Moreover, the positive, as well as significant coefficients associated with the lagged dependent variable, confirm the persistence of FP, which relies on its previous insights. Table 3 shows that there were

coefficients pertaining to the lagged dependent variables for the GMM model, between the fixed effects model and the coefficients of the pooled OLS. These results are comparable to those that were presented earlier (Blundell et al. 2001; Bond et al. 2001), in which the authors achieved an effective and unbiased system GMM. The results were found to be encouraging, which aided us to apply the GMM two-step estimator. Moreover, the reduced-form of the partial adjustment FP model can be determined by employing this two-step system-GMM estimator. The system-GMM method allowed removing time-invariant, unobservable and firm-specific impacts and it could be employed for estimating the initial differences pertaining to each underlying variable. This also allowed managing the correlation between the regressors and residuals. This system allowed minimizing the likelihood regarding endogeneity as it made use of differential equations by employing lagged levels of variables or equations, along with the lagged values pertaining to the first-differences for the variables.

It was observed that all the impacts were cast by the control variables on the FP. Based on the firm's size (i.e., total asset), a negative effect on the FP was confirmed. It was also observed that the ROA was negatively impacted by the advertising ratio. The ROA was positively influenced by the revenue and free cash. A positive relationship was observed between the ROA and leverage. This effect was found to be ambivalent, which could be an increase in the debt levels and was likely to cast a positive impact on the FP caused by an increase in the interest expenditure, which led to further increase in financing costs as part of company's strategies (Hall and Weiss 1967). However, a positive impact was shown by the debt, which mitigates issues of the agency, as over-investment of free cash-flow is strongly discouraged, employed by self-serving managers (Stulz 1990; Harvey et al. 2004).

**Table 3.** The Impact of CPA on Firm Performance (ROA).

| | Static | | Dynamic | | |
|---|---|---|---|---|---|
| **Variables** | **Pooled OLS** | **Fixed Effect** | **Pooled OLS** | **Fixed Effect** | **System GMM** |
| $ROA_{t-1}$ | | | 0.563 *** | 0.245 *** | 0.278 *** |
| | | | (0.0355) | (0.0369) | (0.1680) |
| ln CPA | 0.458 *** | 0.178 | 0.234 | −0.298 | −0.209 |
| | (0.0829) | (0.287) | (0.0478) | (0.287) | (0.186) |
| Leverage | −0.00879 * | −0.00471 | −0.00247 | −0.00413 | −0.000831 |
| | (0.00387) | (0.00530) | (0.00506) | (0.00289) | (0.0000889) |
| Free Cash | 0.00846 *** | 0.0184 ** | 0.0234 *** | 0.0212 | 0.0132 |
| | (0.00869) | (0.00787) | (0.00817) | (0.00872) | (0.0221) |
| Advertising | 0.0276 | −0.178 *** | 0.00753 | −0.231 *** | −0.167 |
| | (0.0171) | (0.0265) | (0.00897) | (0.0534) | (0.0966) |
| ln Total Assets | −1.752 *** | −3.124 *** | −0.763 *** | −5.264 *** | −3.245 *** |
| | (0.213) | (0.548) | (0.243) | (0.789) | (1.458) |
| ln Revenue | 1.205 *** | 2.899 *** | 0.356 ** | 6.709 *** | 4.153 ** |
| | (0.291) | (0.641) | (0.245) | (0.763) | (1.745) |
| Year Dummy | Yes | Yes | Yes | Yes | Yes |
| Constant | 4.891 ** | 12.879 ** | 1.874 | −11.453 * | 58.642 |
| | (1.924) | (5.789) | (1.991) | (6.324) | (10.841) |
| Observation | 1000 | 1000 | 1000 | 1000 | 1000 |
| Number of Firms | | 100 | | 100 | 100 |
| Number of Instruments | | | | | 22 |
| AR(1) | | | | | −3.17 (0.001) |
| AR(2) | | | | | −0.14 (0.756) |
| Hansen Test | | | | | 48.37 (0.178) |
| Different-in-Hansen Test | | | | | 6.45 (0.265) |

Notes: The standard errors are reported in parentheses, except for Hansen test, AR(1), AR(2) and Difference-in-Hansen which are *p*-values. ***, ** and * indicate significance at 1%, 5% and 10% levels, respectively. Time dummies are included in the model specification, but the results are not reported to save space. System GMM model is estimated by using the Blundell and Bond (1998) dynamic panel system GMM estimations and Roodman (2009)—Stata xtabond2 command.

The estimates for CPA rely on the applied model. Few biases could be seen when overlooking the unobservable heterogeneity (by employing the pooled OLS model) as well as the relationship with the dynamic CPA-FP (with regards to the fixed-effects model). For example, a positive relationship between FP and CPA was seen with the System GMM approach as well as the estimates for pooled OLS. However, a neutral relationship was observed with the fixed-effects model. Hence, the dynamics need to be considered when evaluating the relationship for CPA-FP. The pooled OLS estimates and the fixed-effects model were employed to determine the biased results and hence all comments were focused on the GMM findings. Table 3 presents the estimates pertaining to the built model and the effect cast by CPA on FP, which is part of WMAC. The results (with the Dynamic System GMM model, Table 3) showed that FP was not impacted by CPA. The CPA's coefficient was both insignificant and negative, signifying that the firm's CPA does not improve FP. These data support H1 and specify that CPA does not have a direct impact on FP. Based on this, the study verified that a neutral relationship exists between firm's CPA activity and financial performance, which is in line with the earlier empirical literature (Ansolabehere et al. 2004; Hersch et al. 2008).

*5.2. Determining the Interactive Effect of the CP and CPA on the FP*

All results pertaining to the regression model employed for testing the study's Hypothesis 2 are shown in Table 4. In Model 3, the interactive impacts were presented to evaluate all conjectured boundary conditions that were employed in the study. Since Model 3 was a qualified model, it was employed to test Hypothesis 2 as well as demonstrate the variable's effect with accuracy. According to Hypothesis 2, CP was predicted to be a positive moderator that allowed establishing a positive relationship between the CPA and FP. As per this study, the interaction between CP and CPA was significant, even though it was negative ($\beta = -1.044$, $p$-value $< 0.05$). This resulted in the rejection of Hypothesis 2. Also, the results presented that if a high level of CP of the firm, it results in weakening of the CPA impact on ROA, which also results in the rejection of Hypothesis 2. However, better returns could be shown by the low level of CP firms that applied CPA compare to firms displaying a high CP.

Under such conditions, if contradictory CPA activities are shown by a firm, misalignment between CP and CPA occurs. Based on the company's character mechanism that allows reputational change, this study suggested that misalignment is considered in a negative way by corporate constituencies (Love and Kraatz 2009). If there is a misalignment between the company's CP and CPA, stakeholders consider the company to be misleading intentionally, which also makes the company's trustworthiness doubtful. Due to this, the perception of the evaluative audience regarding the company's reputation may change, which has a negative impact on its profit. CP position is mentioned by many companies during their corporate communication, which supported the industry in which they are being lobbied with regards to their conventional business interests.

A good case scenario was the Global Climate Coalition (GCC) (Levy and Egan 2003). In 1989, the GCC was established and supported by many users and producers of fossil fuels, such as car manufacturers and oil or other energy-intensive companies. The GCC embodied the combined interests of all organizations involved in the US climate change as well as for other energy-related policies. For example, the GCC would lobby the Congress to stop the execution of certain regulatory measures (Kolk and Levy 2001; Levy and Egan 2003). During the 1990s, two GCC members, that is, Shell and BP which are recognized as one of the most reputable companies in the world, started an initiative to apply and formulate firm CSR guidelines, producing environmental reports and investing in alternative and renewable energy sources to enhance their reputation. Due to this, the GCC's public position started becoming irreconcilable, mainly due to activist campaigns. Hence, Shell and BP had to leave this coalition. In 2002, complete dissolution of the GCC occurred, a post which various companies started building better business alliances with other companies supporting climate change policies, such as the US Climate Action Partnership and the World Business Council for Sustainable Development. For certain strategic circumstances, firms can deliberately seek misalignment and henceforth remain associated with CPA and CP. However, this factor may also arise over a time period, for example, if

the company is unable to maintain its CP and CPA coordination or if some dynamics cast an impact on the companies' initial alignment with respect to their corresponding fields.

**Table 4.** The Moderating Effect of CP on CPA and Firm Performance (ROA).

| | Dynamic | | |
|---|---|---|---|
| **Variables** | **Model 1** | **Model 2** | **Model 3** |
| $ROA_{t-1}$ | 0.247 *** (0.0415) | 0.187 *** (0.0229) | 0.212 *** (0.0624) |
| ln CPA | | −0.245 (0.577) | −5.591 ** (2.415) |
| CP*CPA | | | −0.899 ** (0.579) |
| CP | 0.235 (0.263) | 0.475 * (0.345) | 7.598 ** (3.863) |
| Leverage | −0.00247 (0.00506) | −0.00413 (0.00289) | −0.000831 (0.0000889) |
| Free Cash | −0.000897 (0.00144) | −0.000458 (0.00143) | −0.000558 (0.000331) |
| Advertising | −0.223 *** (0.0504) | −0.241 *** (0.0507) | −0.351 *** (0.0578) |
| ln Total Assets | −2.863 *** (0.443) | −2.254 *** (0.669) | −5.265 *** (1.878) |
| ln Revenue | 4.356 *** (0.875) | 5.709 *** (0.633) | 4.153 ** (1.885) |
| Year Dummy | Yes | Yes | Yes |
| Constant | 34.878 (5.981) | 28.455 (4.366) | −48.645 (11.551) |
| Observation | 1000 | 1000 | 1000 |
| Number of Firms | 100 | 100 | 100 |
| Number of Instruments | 26 | 26 | 26 |
| AR(1) | −3.45 (0.001) | −3.15 (0.003) | −3.18 (0.001) |
| AR(2) | 0.23 (0.879) | 0.55 (0.468) | −0.534 (0.856) |
| Hansen Test | 23.08 (0.325) | 40.33 (0.423) | 34.37 (0.658) |
| Different-in-Hansen Test | 7.37 (0.266) | 2.74 (0.696) | 5.45 (0.345) |

Notes: The standard errors are reported in parentheses, except for Hansen test, AR(1), AR(2) and Difference-in-Hansen which are *p*-values. ***, ** and * indicate significance at 1%, 5% and 10% levels, respectively. Time dummies are included in the model specification, but the results are not reported to save space. System GMM model is estimated by using the Blundell and Bond (1998) dynamic panel system GMM estimations and Roodman (2009)—Stata xtabond2 command.

Based on the results obtained by employing Table 4—Model 3, Aiken et al. (1991) method was employed to show better moderating impacts as well as to plot the significant interaction effects (at $p < 0.10$ or greater). A standard deviation was seen either below or above the mean corresponding to moderating variables' low and high levels, respectively. Since a significant interaction was seen between CP and CPA, it was considered helpful for further visual exploration. Figure 2 shows the impact cast by CPA on ROA pertaining to two distinct CP levels (low and high). With regards to the moderation impact, Figure 2 demonstrates that a stronger impact is cast by CPA on ROA for low CP levels. High CP levels signify a weaker association to be present between FP and CPA. Here, it was verified that CP considerably moderates the impact cast by CPA on FP. In other words, companies with high reputation are associated with a stronger negative relationship between FP and CPA. However, this interaction pattern is not consistent with the study's expectation, concluding that Hypothesis 2 was not supported.

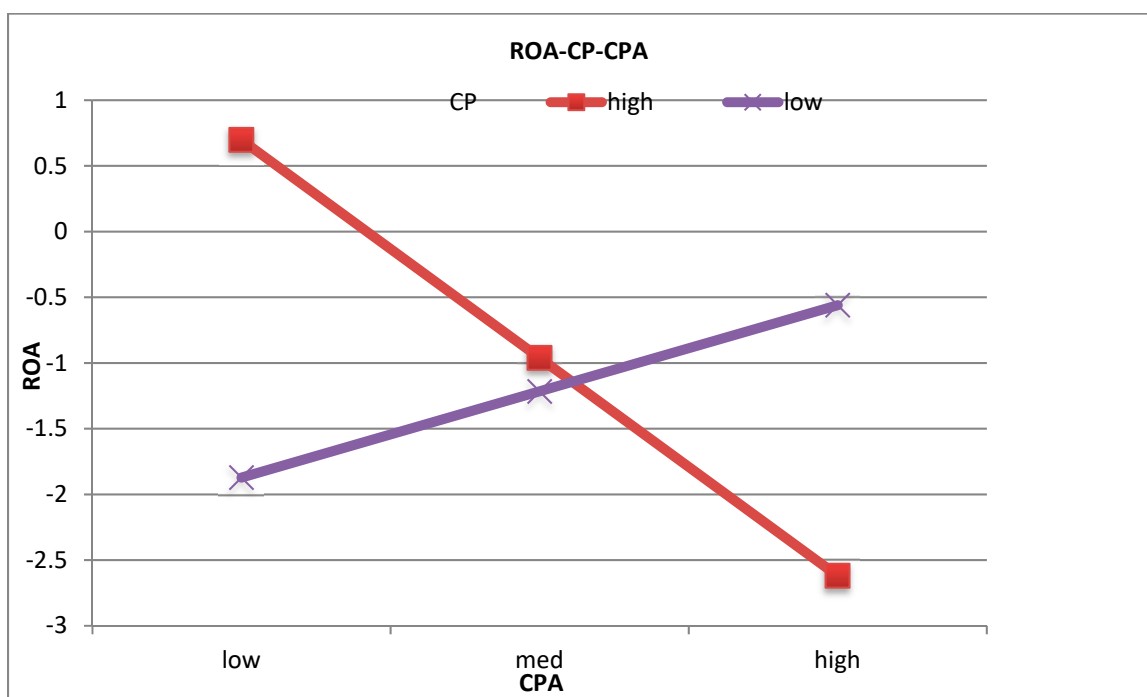

**Figure 2.** Effects of CPA on ROA: Contingent on CP.

A possible explanation could be when CPA is confronted by stakeholders or individuals, such as lobbying, since they normally have a bad perception regarding this. The critics could sometimes be too harsh making lobbyists think that they had best stayed in the shadows. The term lobbying has now earned a bad reputation and is normally associated with words like corruption, manipulation, bribery and so forth. The public hear of lobbying in the media. Crisis exists everywhere and scandals sell rather quickly, which is the reason the media pick up on bad stories first, involving lobbyists, instead of good ones. In that context, it seems logical that lobbying continues to be a 'dark activity' and that lobbyists are, in general, considered as bad guys. Thus, when the firm's CPA is high, an inverse effect is seen with the reputable company.

## 6. Robustness Test

ROE was used as an alternative measure for FP and a robustness check was performed. By keeping the basis as a performance-dependent variable, which coincides with, who from a viewpoint of operations research stressed that "more enhanced assessments regarding the dependability and validity of the construct can be achieved with a comprehensive, multifaceted performance measurement, which increases the confidence in the findings." As per a firm's FP can be measured distinctly by employing return on assets (ROA). With regards to robustness, an additional criterion was ROE. DataStream was employed to derive data on ROE. In Panel A of Tables 3 and 4 and Figure 2, the presented analyses were repeated again by employing the same approach for more robustness tests. Thus, instead of ROA, ROE was employed to test the study hypotheses. The results are presented in Tables 5 and 6 and Figure 3. Moreover, the results were similar to the results presented in Tables 3 and 4 and Figure 2. There was no significant difference either insignificance or in coefficients signs. By employing post-estimation for each robustness check, based on the diagnostic tests, well-specified models are recommended for all. The over-identification restrictions were not rejected by the Hansen test, while for the difference-in-Hansen test, rejection could be seen. The study also allowed identifying a lack of second-order autocorrelation AR(2). The number of cross-section firms was found to be greater than the number of instruments, which was deemed acceptable. Further, the study did not see any instrument proliferation issues. Therefore, the study can be employed to claim robust empirical results.

**Table 5.** The Impact of CPA and Firm performance (ROE).

| Variables | Static | | Dynamic | | |
| | Pooled OLS | Fixed Effect | Pooled OLS | Fixed Effect | System GMM |
|---|---|---|---|---|---|
| $ROE_{t-1}$ | | | 0.574 *** (0.0245) | 0.267 *** (0.0289) | 0.348 *** (0.0880) |
| ln CPA | 2.448 *** (0.599) | 0.413 (1.267) | 0.785 ** (0. 478) | −1.265 (1.402) | 0.0578 (0.186) |
| Leverage | 0.608 *** (0.0287) | 0.306 *** (0.0530) | −0.347 *** (0.0406) | −0.413 *** (0.0209) | −0.351 *** (0.0789) |
| Free Cash | 0.786 *** (0.0569) | 0.0564 (0.0530) | 0.0434 *** (0.0417) | 0.0241 (0.0572) | −0.0132 (0.321) |
| Advertising | 0.0287 (0.0371) | −0.778 *** (0.215) | −0.00402 (0.00567) | −0.731 *** (0.245) | −0.167 * (0.156) |
| ln Total Assets | −5.652 *** (0.913) | −13.324 *** (0.848) | −3.763 *** (0.663) | −5.286 *** (0.889) | −10.245 (4.458) |
| ln Revenue | 2.305 *** (0.411) | 2.339 *** (0.751) | 0.456 ** (0.345) | 6.719 *** (0.233) | 4.153 (1.545) |
| Year Dummy | Yes | Yes | Yes | Yes | Yes |
| Constant | −5.661 ** (1.874) | 4.879 ** (6.749) | 3.874 (1.131) | −9.453 * (6.424) | 5.642 (1.821) |
| Observation | 1000 | 1000 | 1000 | 1000 | 1000 |
| Number of Firms | | 100 | | 100 | 100 |
| Number of Instruments | | | | | 22 |
| AR(1) | | | | | −3.26 (0.021) |
| AR(2) | | | | | 1.84 (0.356) |
| Hansen Test | | | | | 53.37 (0.138) |
| Different-in-Hansen Test | | | | | 4.45 (0.165) |

Notes: The standard errors are reported in parentheses, except for Hansen test, AR(1), AR(2) and Difference-in-Hansen which are *p*-values. ***, ** and * indicate significance at 1%, 5% and 10% levels, respectively. Time dummies are included in the model specification, but the results are not reported to save space. System GMM model is estimated by using the Blundell and Bond (1998) dynamic panel system GMM estimations and Roodman (2009)—Stata xtabond2 command.

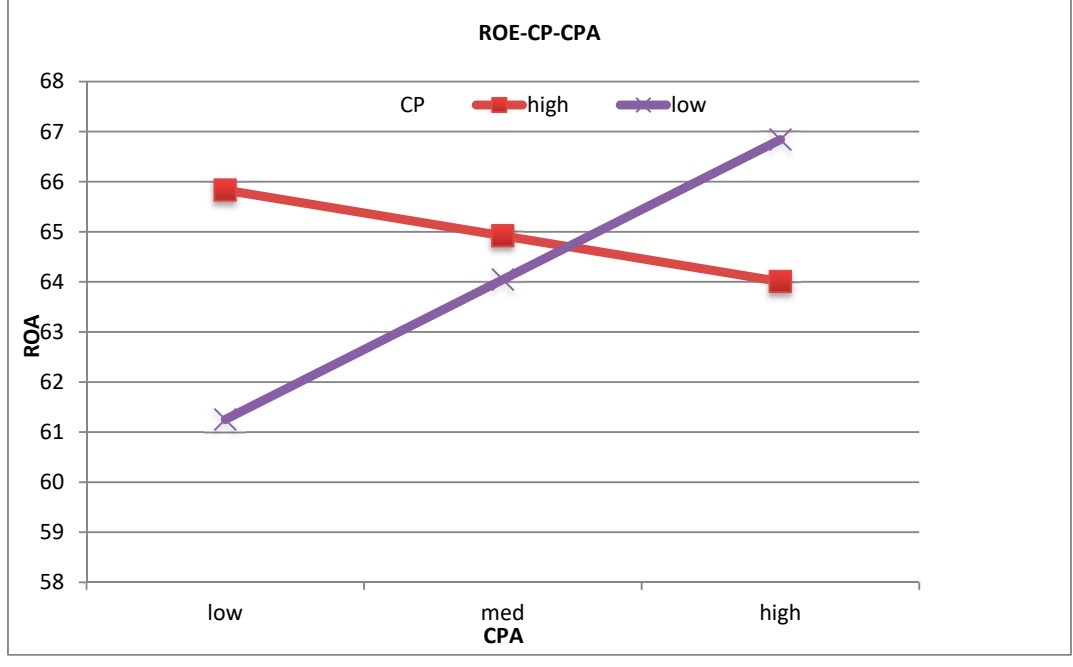

**Figure 3.** Effects of CPA on ROE: Contingent on CP.

**Table 6.** The Moderating Effect of CP on CPA and Firm Performance (ROE).

| | Dynamic | | |
|---|---|---|---|
| **Variables** | **Model 1** | **Model 2** | **Model 3** |
| $ROE_{t-1}$ | 0.327 *** (0.00455) | 0.387 *** (0.00429) | 0.289 *** (0.0124) |
| ln CPA | | −0.233 (0.427) | −6.531 *** (0.615) |
| CP*CPA | | | −0.877 *** (0.899) |
| CP | 0.455 (0.243) | 1.475 * (0.445) | 5.598 ** (0.893) |
| Leverage | 0.237 *** (0.00406) | 0.423 *** (0.00349) | 0.731 *** (0.00759) |
| Free Cash | −0.0297 (0.0644) | −0.0558 (0.0343) | −0.0358 (0.0431) |
| Advertising | −0.403 *** (0.1104) | −0.751 *** (0.3507) | −0.341 *** (0.0478) |
| ln Total Assets | −4.763 *** (0.533) | −2.789 *** (0.579) | −5.532 *** (1.228) |
| ln Revenue | 5.356 (0.445) | 5.219 (0.433) | 3.153 * (1.855) |
| Year Dummy | Yes | Yes | Yes |
| Constant | 29.898 (3.981) | 18.456 (3.366) | -28.345 (1.451) |
| Observation | 1000 | 1000 | 1000 |
| Number of Firms | | 100 | 100 |
| Number of Instruments | 26 | 26 | 26 |
| AR(1) | −2.45 (0.011) | −3.25 (0.002) | −3.38 (0.000) |
| AR(2) | 0.33 (0.869) | 0.55 (0.488) | −0.434 (0.846) |
| Hansen Test | 13.08 (0.625) | 10.13 (0.523) | 14.37 (0.768) |
| Different-in-Hansen Test | 2.37 (0.366) | 2.44 (0.696) | 3.45 (0.345) |

Notes: The standard errors are reported in parentheses, except for Hansen test, AR(1), AR(2) and Difference-in-Hansen which are *p*-values. ∗∗∗, ∗∗ and ∗ indicate significance at 1%, 5% and 10% levels, respectively. Time dummies are included in the model specification, but the results are not reported to save space. System GMM model is estimated by using the Blundell and Bond (1998) dynamic panel system GMM estimations and Roodman (2009)—Stata xtabond2 command.

## 7. Conclusions

### 7.1. Implications for Theory and Literature

In this study, this study made an effort to provide numerous positive suggestions to aid in future research, which can be implemented in the society even though this was based on secondary data. Moreover, different stakeholders were able to gain useful insights. Also, this report investigates the complicated relationship existing between FP and CPA, when CP is present. This study employed the WMAC data from the period 2007 to 2016, which showed that there was an insignificant relationship between FP and CPA. First, the contribution of this report to the literature is in accordance with the political activities. The study results showed that the firms, based on their non-market activities, failed to gain a competitive advantage and CPA is an agency cost to the firm. Lobbying and PACs seem to provide limited tangible benefits in terms of helping a firm obtain government contracts or successfully pass the congressional bills lobbied. These results suggest that agency costs dominate the strategic benefits of political activities, particularly when public policies are considered. Despite the earlier evidence explaining this relationship (Ansolabehere et al. 2004; Hersch et al. 2008), there are

still numerous contrasting views predominant in terms of FP and CPA (Hillman 2005; Bonardi et al. 2006; Richter et al. 2009). Though the study results support the work of Ansolabehere et al. (2003), they could also offer evidence that supports the arguments. As described earlier, this study is an attempt to contribute to the existing literature as well as theoretical contributions to an extent. There are very few studies that employ the CPA. Moreover, the CPA-FP relationship was tested with a lack of empirical study (Wesseling et al. 2015). CPA, as well as other factors associated with CPA, is a very crucial parameter as various profit-seeking companies are constantly trying to maximize their profits or improve their market shares through contact with public policy makers instead of applying good, competitive market-based measures.

### 7.2. Managerial and Policy Implications

The study's empirical results were employed to derive numerous practical implications. However, depending on the individual's perspective, these implications could also contradict each other. For instance, with regards to the managerial viewpoint, the study's results do not motivate firms to transfer resources to the non-market activities (i.e., public policy intervention) from their market activities (i.e., better services or products), since such type of actions does not result in any profitable outcome. However, it is difficult for corporate decision-makers to derive a subtle perspective for these implications. This is because blind spending of resources is not propagated by the results. While employing public policies increases the firm's internal capabilities, the management can also benefit. On the surface, the CPA is seen as a networking transaction. But, on the contrary, firms with optimized political connections were observed to remain deeply committed towards organizational division development, which involved maintaining proper connections based on the objectives of the organization. This aids companies to get involved in better political lobbying since proper political connections allow gaining knowledge as well as determine proper means to improve political spending appropriately (instead of spending blindly) (Kim et al. 2013). On viewing this situation from the viewpoint of a policy, opposing implications could be seen. For instance, if there is a continuous talent drain from public companies to private firms, both are at loss as both face difficulties in meeting their organizational objectives. Public companies may also possibly face the risk of regulatory take-over from these private firms. Hence, the government offices should pay attention to mitigating all risks to preserve their transparent legal processes, both in terms of their administrative and legislative efforts.

### 7.3. Future Research

Going forward, researchers can deploy various kinds of techniques for enhancing the literature on the subject. For this, a qualitative investigation aimed at the relationship between CPA and FP could be performed. For instance, a multiple case study could be conducted for specific industries such as the air transport or the mining industries, for enhancing the basic CPA literature or, more precisely, the air transport/mining-related CPA literature. Even though this kind of study is considered to be more empirical and theoretical, a thorough study has not been performed. Secondly, there is a lack of studies investigating the antecedents pertaining to the CPA. This study should note that a lot of efforts are put by companies to design public policies, which allows gaining a proper understanding of the variation existing amongst several industries. Thus, various firm-level CPAs can be determined, which also helps to contribute to the existing literature about CPA.

**Funding:** The research was funded by Universiti Putra Malaysia, grant number GP-IPS 9536600.

**Conflicts of Interest:** The author declares no conflict of interest.

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
