# Peer review of "Is Corporate Political Activity an Investment or Agency? An Application of System GMM Approach"

_admsci, doi:10.3390/admsci9010005_

Round 1

Reviewer 1 Report

See attached referee report.

Author Response

To dearest Mr Reviewer, 

I am very much thankful to the reviewers for their deep and thorough review. I have revised my present research paper in the light of their useful suggestions and comments. I hope my revision has improved the paper to a level of their satisfaction.  Number wise answers to their specific comments/suggestions/queries are as follows. (please refer file attached) 

Reviewer 2 Report

This is a really interesting study overall, and I applaud your efforts to try to make some headway using quantitative analysis to make your point.  I think it would and could be a better study if you had focused exclusively on CSR instead of CPA, however, and I also think that your proxy for CSR is not as strong as it could be.  The Most Admired is only partially connected to CSR.  Instead, you should have tried to use a meta analysis of all CSR measurements and use only the portion of the most admired that is actually focused on CSR.  As a result, I think the outcome of your work is flawed, but fascinating.  If it helps others to think of ways to measure these things, I applaud your work.  But you just might want to rethink what and how you are measuring these things.  I also think a stronger opening about what your are trying to do would help, and I would rely more heavily on the work of Porter and Kramer for the connection between CSR and financial performance.

Author Response

To dearest Mr Reviewer, 

I am very much thankful to the reviewers for their deep and thorough review. I have revised my present research paper in the light of their useful suggestions and comments. I hope my revision has improved the paper to a level of their satisfaction.  Number wise answers to their specific comments/suggestions/queries are as follows. (please refer to attachment) 
